# Consumer Preference for Attributes of Single-Use and Multi-Use Plastic Shopping Bags in Cape Town: A Choice Experiment Approach

Victor Virimai Mugobo *⬛ and Herbert Ntuli ⬛

Department of Retail Business Management, Faculty of Business and Management Sciences, Cape Peninsula University of Technology, Cape Town 7530, South Africa
* Correspondence: mugobov@cput.ac.za

**Abstract:** Single-use plastic bags are increasingly becoming unpopular across the globe due to growing concerns over plastic pollution, which is threatening both terrestrial and marine ecosystems. Creating policy interventions to reduce plastic consumption requires objective information about local conditions. This study uses a choice experiment to elicit consumer preference for attributes of shopping bags from a sample of 250 consumers in Cape Town. Following the literature, we estimate the conditional logit model and the mix logit model and perform appropriate tests to establish a model which fits our data. Our results show that consumers in Cape Town prefer small and medium-sized bags relative to the status quo. They also care about durability, reusability, style, and recyclability of shopping bags. The highest willingness to pay is associated with a medium-sized shopping bag (R3.76), followed by a shopping bag that is reusable (R3.35), then a shopping bag that is styled (R2.16), then a small-sized shopping bag (R1.74), then a durable shopping bag (R1.50) and finally a recyclable shopping bag (R1.25). By aggregation, the willingness to pay of a small-sized shopping bag that is recyclable is R2.99 which is equivalent to the maximum price that the respondent is offering for a single-use plastic bag (R2.92). The willingness to pay of a medium-sized and reusable shopping bag is R7.11 per unit which is at least as high as the minimum price that respondents would consider for alternative packaging (R7.37). Finally, taking into consideration all the important attributes, we arrive at a willingness to pay ranging from R9.99 to R12.01 per bag for a small and medium shopping bag, respectively. Our results call for a combination of policy instruments such as a subsidy on expensive durable and reusable shopping bags to increase demand while at the same time increasing the levy on single-use plastic bags to reduce demand. More effort is needed to increase appreciation and perception of recyclable products.

**Keywords:** single-use; multi-use; plastic shopping bags; choice experiment; Cape Town

## 1. Introduction

Plastic pollution is increasingly gaining attention from researchers and policymakers at the international level since it is a public good with a global dimension [1–6]. In particular, the use of single-use plastic bags has been identified by many scholars as one of the major sources of plastic pollution, especially those with inherent chemical properties which make them non-biodegradable [4,7]. In addition to this undesirable attribute, other characteristics such as cheapness, convenience and availability make single-use plastic bags more likely to be pushed by retailers at the expense of durable alternatives on the market [8]. While the ecosystem lacks the ability to sequester such type of waste material generated by economic activities, the consumption of non-biodegradable plastics imposes an externality on society by compromising environmental aesthetics or beauty which translate into reduced market value to real estate properties [9], posing harm to both terrestrial and marine wildlife [10], and blocking the drainage system in cities, thereby exacerbating the incidences of diseases

and floods [11]. Although the evidence is still scarce, microplastics are also believed to cause health problems in human beings [12].

Globally, seventy million tons of plastics are generated annually from economic activities associated with the production, distribution, and consumption of household goods [6]. Thousands of people depend on the plastic industry as a source of livelihood across the globe being employed in either a formal or informal plastic value chain [3,9]. Evidence reveals that plastic consumption is on the rise, especially in developing countries due to rapid urbanization caused by high population growth rates, rural–urban migration and rising household incomes resulting from long working hours and increased participation of women in the labour force [13]. Projections show that the population in developing countries is expected to double by 2050 being attributed to growth in the share of informal settlements or urban slums [3], which in turn might translate into increased plastic consumption in the absence of appropriate measures [14].

As a matter of concern, plastic pollution is increasing despite efforts by policymakers to reduce consumption through policy interventions ranging from simple behavioural interventions such as information provisioning or awareness campaigns [3] to market-based instruments such as plastic levies [2]. This situation culminated in most governments across the globe (including a few leaders in developing countries in Africa such as Kenya, Uganda, and Rwanda) implementing hush policies in the form of a ban on single-use plastic packaging [15]. Such a policy is often difficult to justify given the contribution of the plastic sector to the whole economy and the fear that the majority of households depending on the plastic industry for survival will end up losing their jobs [13,16]. As a result, this policy has been highly contested in Southern African countries such as South Africa and Zimbabwe [15,17].

Policy instruments in developing countries are fervently endorsed based on weak empirical evidence and as a result, such policies fail to achieve the desired outcomes [17]. For instance, little is known about consumer preferences for attributes of shopping bags such as the convenience of use or disposal, recyclable, reusable, durability, and style. Important stakeholders such as consumers affected by a government policy are not usually consulted during policy designs and studies of this nature allow actor or beneficiaries to communicate their needs and wants [18]. Based on a review of empirical literature from developing countries, ref. [8] observed some gaps with regard to information about why and how consumers choose among the different types of shopping bags, be they plastic (single-use plastic bags versus recyclable), paper or cloth. This important information is part of the building blocks in understanding consumer behaviour and can help policymakers to craft better policies aimed at addressing plastic pollution. Furthermore, most of the previous studies were either qualitative in nature or employed descriptive statistics (e.g., [9,16,19–21], while few studies employed rigorous econometric techniques to provide objective evidence relevant for policymaking [22,23]. As a result, most of the policies used in African countries to curb plastic consumption were adopted from first world countries with little or no effort made to adapt the instruments to suit local conditions.

Given the background above, three important policy questions arise: (i) Which attributes matter for purchase decisions involving alternative shopping bags available on the market in Cape Town? (ii) Does the choice of shopping bags and their attributes tell us something about sustainable behaviour? (iii) How can policy facilitate the adoption of sustainable behaviour by influencing consumer choice among the alternative options? This study uses a choice experiment to elicit consumer preference for attributes of shopping bags in the context of an African city such as Cape Town where plastic consumption is increasingly being exacerbated by rapid urbanization coupled with a high population growth rate. By so doing, our study provides pragmatic evidence to policymakers and insights into consumer tastes and preferences for alternative packaging that is available on the market. While consumers are usually able to communicate their tastes and preferences for most products within budget through actual purchase, preference for some products outside the consumer's reach remains largely unobservable. An understanding of the

attributes that influence the consumer's decision to buy or choose a particular type of shopping bag over another is required to craft better policy interventions to incentivize sustainable behaviour so that potential demand for alternative options that are ordinarily not considered in the consumer's shopping basket for some reason can be translated into effective demand in the future. Our study also contributes to the literature addressing plastic pollution through modelling consumer preferences which indirectly speaks to the adoption of sustainable behaviour either by reducing the consumption of single-use plastic bags or increasing the demand for durable multi-use packaging. We also contribute to the literature on the methodological front.

The rest of the paper is organized as follows. Section 2 presents the design of the choice experiment paying particular attention to the procedure used in its design, including a detailed description of the attributes, while Section 3 describes the case study. Section 4 focuses on the research methods, theoretical underpinnings of the model, sampling and data. The results and discussion of this study are presented in Sections 5 and 6, respectively. Finally, we conclude and provided the policy recommendations in Section 7.

## 2. Design of the Choice Experiment

### 2.1. Procedure Used in Designing the Choice Experiment

In this section, we address the methodology followed in designing the choice experiment (CE) used in this study paying particular attention to the principles laid out in [24]. According to [24,25] the primary goal of experimental design in CEs is to develop designs that yield efficient and unbiased estimates of preference parameters and value estimates. In defining policy-relevant attributes for a wide range of shopping bags and levels, we conducted a qualitative review of the existing literature. Following the CE literature, the attributes and levels were then refined using additional information obtained from a pilot study, focus group discussions (FGDs) with consumers from six different locations selected according to income (low, medium and high) each with an average of 8 participants and expert opinions based on key informant interviews with 3 authorities from the city of Cape Town and 6 supermarket managers from the selected locations [18,24,26,27]. The locations and key informants were purposefully selected based on the researcher's knowledge. Snowball sampling was used to identify other key informants, while the FGD participants were recruited with assistance from the shop managers. Both FGDs and key informant interviews were also used to gather qualitative information that was later on used to fill in the gaps in quantitative analysis. During the pilot study, we conducted face-to-face interviews in four suburbs in Cape Town with similar characteristics to the 13 suburbs sampled in the study stratified according to their social standing, i.e., very high-income, high-income, medium-income and low-income areas. The purpose of the pilot study was to refine the attributes and make sure that they are understandable by respondents in different categories. Three in-depth interviews were conducted in each of the four suburbs included in the pilot study to gather qualitative information that will be used to support our quantitative analysis. Through these in-depth interviews, we were able to develop a localized understanding of important concepts associated with identified attributes and a way to convey them to the respondents.

The attributes were conveyed in picture form so that respondents were able to relate what they know to the questions being asked [18,26,28]. In a study to examine the effects of presentation formats in choice experiments, namely text, visuals, and a combination of both, ref. [26] demonstrated that the visual format generates more statistically significant coefficients than the other formats, suggesting that the presentation format has significant impacts on choice. According to [18], he pictorial presentation minimizes biases related to the level of education of respondents. Furthermore, concepts and terms are interpreted differently by respondents from different backgrounds, and convergence in terms of interpretation is likely to occur if a picture is provided [29]. Pictures also minimize the use of too many words to explain a concept which might discourage the respondent [26]. Stated

preference (SP) studies should elicit evidence of information pieces that are understood, accepted, and viewed as credible by respondents [24].

To elicit consumer preferences for shopping bag attributes, the study used an internet-based survey CE. The picture format allowed the attributes and message to be conveyed in a way that respondents understood [10,24,26]. To make sure that it complies with ethical standards, the survey instrument was evaluated by colleagues in the Department of Retail Business Management and the Ethics Committee of the Faculty of Business and Management Sciences at the Cape Peninsular University of Technology. The survey had various sections on the socio-economic characteristics of the respondent, consumer awareness and perceptions of alternative shopping bags, their buying behaviour and the alternative policy scenarios including the baseline. In this survey, respondents are presented with a series of choice alternatives, differing in terms of the levels of shopping bag attributes, and asked to choose their most preferred options among a range of alternatives presented to them.

Arguably, one of the most important attributes of the CE model is cost. Theoretically, we expect the cost to carry a negative sign when regressed against the choices made by respondents (used as the dependent variable in the model) for us to be able to operate in the CE space. In addition to the costs, the important attributes identified in this study are convenience of use, degradability, recyclability, reusability, durability, and style (including branding). These attributes are associated with shopping bags available in Cape Town on the market such as nondegradable single-use plastic bags, degradable paper bags, recyclable plastic bags and durable bags such as taxi bags and bags made of cloth. Another variable that we include in the model is the non-status quo alternatives (ASC) which is a measure of bias [18]. This variable takes a value of 1 for the status quo and 0 otherwise. The status quo is defined as what the customers-cum-respondents are currently using and the type of shopping bag that they are using.

Based on the results of the pilot study, we decided to drop degradability since the concept was difficult to convey, respondents could not differentiate between degradable and non-degradable packaging and showing pictures of plastic pollution would influence the results. The attributes that we included in the model are those that are either communicated to customers through labelling or experienced by a customer during shopping. Furthermore, the non-degradability of plastic bags is an attribute that matters after purchase and is associated with the pollution which enters the society's welfare function as an externality [20,30]. As a result, it is difficult to justify the association between an individual's purchase decision and the non-degradability of plastic bags in our model since they do not incur the costs of pollution. What we include in the CE are attribute that matter during a purchase. We believe that the recyclability, reusability, and durability of shopping bags are neutral attributes to capture the environmental concerns of green consumers. There is an overlap in these attributes in terms of how they are defined and viewed by consumers. The identified attributes and their levels are presented in Table 1, followed by a discussion of what they represent.

### 2.1.1. Convenience

Shopping bags bring convenience to the shopper since they act as packaging and allow the consumer to carry most household goods easily regardless of type [9]. According to [31] the size of a plastic bag is one of the most important features which brings convenience to a customer. Shopping bags come in all sizes and customers make a decision whether to buy a small, medium or large shopping bag depending on their cost, the type of goods they want to carry, distance from home and whether they have access to a car or not.

Small-size shopping bags allow customers to distribute the weight of goods for easy carrying and sorting goods by separating food items from non-food items, especially chemicals which are dangerous if consumed such as detergents, bleaching compounds and insecticides [32]. These chemicals can lead to illness and sometimes death if consumed in excess, depending on the nature of the poison [33]. The authors also observed that some

of the non-food items can also contaminate the food if combined through spillages which affect the taste and smell of food. Even food and non-food items might also need to be separated to avoid contamination, e.g., liquid and non-liquid items.

**Table 1.** Attributes associated with the use of shopping bags (source: literature review).

| Attributes | Level | | |
|---|---|---|---|
| | **1** | **2** | **3** |
| **Convenience/size** | Small  | Medium  | Large  |
| **Recyclable** | Non-recyclable  | Recyclable  | |
| **Reusable** | Non-reusable  | Reusable  | |
| **Durability** | Non-durable  | Durable  | |
| **Branding and style** | Not styled  | Styled  | |
| **Cost** | R0.75        R1.50 | R3.25        R7 | R14        R28 |

However, most small plastic bags are usually less durable which means that they are likely to break down in use leading to losses in the form of breakages [34]. This makes small plastic bags relevant for use as single-use packaging. Evidence shows that most consumers in developing countries prefer single-use plastic bags because their cost is negligible relative to the total purchase which also makes them 'cheap' to dispose of, though this type of convenience is undesirable [30]. Furthermore, small plastic bags compete for storage space that could be used to store other household items [33]. Medium and large shopping bags have their advantages and disadvantages. One of the main advantages of relatively large shopping bags is the ability to carry all purchased goods in one bag. This becomes imperative if the types of goods are in a similar category, e.g., either food items or non-food items. Another important advantage of large shopping bags is the ability to save money since they are relatively strong and can be reused multiple times without losing their shape or appeal [9]. Whether customers think in terms of saving money when choosing among small, medium, and large shopping bags is debatable as this depends on the price differences.

### 2.1.2. Recyclability

Another important attribute is the recyclability of shopping bags, especially those that are made of a non-degradable plastic material. Recyclability implies that the material can be used over and over again indefinitely, but the reverse might not be necessarily true [35]. Plastics can be recycled back into plastic bags or other products useful for household consumption or in the manufacturing of other products [9]. Recycling is limited by the cost of production and availability of electricity since the industry depends heavily on energy [36]. Without enough energy sources, the cost of recyclable products exceeds the cost of manufacturing new products from first principles using raw materials since the technology is cheap.

Awareness of recycling initiatives and increased consumer participation in recycling projects are critical for their survival [37]. The consumer's recycling behaviour is believed to be governed by their perceptions and attitudes toward recyclable products and the environment [13]. Theory predicts that consumers care more about the environment as income increases, which means recycling initiatives perform better in relatively wealthier than in poor societies [36]. There is substantial evidence from first world countries demonstrating that environmental concerns translate into WTP for recycling projects [37]. The WTP for such an initiative provides vital information to policymakers in Africa about consumer support and whether consumers are ready and the potential of such projects to succeed.

### 2.1.3. Reusability

There are different types of reusable shopping bags ranging from durable plastic bags to bags that are available on the market ranging from plastic bags to bags made of cloth [32]. While the former does not require maintenance, the latter can be washed and ironed before being stored or used again [38]. Consumers care about the reusability of shopping bags from an environmental point of view, the ability to save money by reducing long-run costs associated with their purchase and as a tool to use for purposes other than shopping [33]. The price of both reusable and non-reusable (single-use) shopping bags matters during a purchase decision. If customers are able to save money by purchasing reusable shopping bags, then the demand for these bags will increase while the demand for single-use plastic bags diminishes. A huge gap associated with the price differentials reduces the money that can be saved which, in turn, reduces the benefits realized by adopting reusable shopping bags [39].

While the durability of shopping bags implies reusability, the reverse is not necessarily true again since some shopping bags that are reusable may not be durable in the eyes of the consumer [34]. According to [9], the length or period over which a shopping bag is reusable also matters in addition to the cost. In a sense, a durable shopping bag can be used several times before reaching its breaking point or limit. However, how many times is a debatable question? Both reusability and durability have not been clearly defined in the literature to reach conclusions about their implications on environmental sustainability in the use of plastic bags. For example, what, when, where and how many reusable or durable plastic bags are required by the circular economy to be able to say that there is a reduction in plastic pollution?

Another important concept related reusability of plastic shopping bags is upcycling. Upcycling, also known as creative reuse, is the process of transforming by-products, waste materials, useless, or unwanted products into new materials or products perceived to be of greater quality, such as artistic value or environmental value [40]. It represents a variety of processes by which "old" products get to be modified and get a second life as they are turned into "new" products. As opposed to reusing plastics, upcycling involved creativity, value addition and an opportunity to generate income through selling the new high-value goods produced from the old plastic products [41]. However, upcycling is not common with plastic bags in South Africa, especially single-use plastic bags. This presents a lot of opportunities to create shoppers and to educate customers about upcycling single-use plastic bags through educational programmes and awareness campaigns.

### 2.1.4. Durability

The need to produce and consume cheap and durable shopping bags should not be underestimated if ever the objective of a clean environment is to be achieved. Current production processes of durable shopping bags emphasize quality, styling and branding which make the product out of reach for poor consumers [32]. The combination of cheapness and durability in shopping bags has received very little attention from both researchers and policymakers in developing countries [8]. How can we make durable shopping bags such as those made from cloth and taxi bags more attractive to poor customers? One way might be to reduce either the production costs of alternative packaging or the cost incurred by the consumer at the point of purchase [5]. The latter could be realized through a subsidy or reduction in tax on durable shopping bags.

### 2.1.5. Style and Branding

For many decades, product styling and branding has been among the most important marketing tool for both manufacturers and retailers. Styling and branding become imperative for durable and reusable shopping bags from a cost point of view. Evidence shows that upmarket retailers have been subsidizing the cost of branding on cheap single-use plastic bags as a marketing tool because the benefits of doing so far outweigh the costs [32]. When it comes to styling, we observe that durable shopping bags are made more attractive, but this obviously comes at a price to the consumer [31]. While shopping bags made of cloth are increasing becoming popular, this type of packaging is preferred by relatively wealthy consumers [33]. Though not superior to plastics, the branding of cloth is meant to leave a lasting impression that lasts and give a sense of identity to the consumer. Although style and branding might influence the demand for shopping bags, the price still remains the most critical factor as most potential consumers are marginalized from the market [31].

### 2.2. Generating Experimental Designs

With 4 attributes varying across two levels each, 1 attribute varying across three levels and 1 attribute varying across six levels, there were $4^2 \times 3^1 \times 6^1$ possible combinations of the attributes and their levels. To minimize bias, a full factorial orthogonal design of 16 alternative profiles was created using NGENE software from the full set of possible combinations. The software produced an efficient design with one status quo and two non-status quo alternatives per choice set, and four choice sets arranged in four survey blocks or cards. If multiple valuation questions of each subject are asked in a CE, then additional trade-offs involving issues such as efficiency, bias, and the evolution of choice heuristics should be considered, and question order should be randomized across respondents [24]. Respondents were randomly assigned one of the four survey blocks which had been prepopulated in six different questionnaire versions. Table 2 provides one example of the CE scenarios that were presented to the respondents.

**Table 2.** Choice experiment scenario (source: own design).

| | BLOCK 1 | | |
|---|---|---|---|
| **Scenario 4** | | | |
| **ATTRIBUTES** | **ALTERNATIVE 1** | **ALTERNATIVE 2** | **STATUS QUO (what you are currently using)** |
| **size** | Small  | Medium  | |

**Table 2.** *Cont.*

| | BLOCK 1 | | |
|---|---|---|---|
| **Scenario 4** | | | |
| **ATTRIBUTES** | **ALTERNATIVE 1** | **ALTERNATIVE 2** | **STATUS QUO** (what you are currently using) |
| **recyclable** | Non-recyclable  | Non-recyclable  | |
| **Reusable** | Non-reusable  | Reusable  | |
| **durability** | Durable  | Non-durable  | |
| **Style** | Not styled  | Not styled  | |
| **Cost** | R3 | R28 | R0 |

## 3. Case Study Area

Cape Town is a multiracial society with consumers from diverse backgrounds and a culture that spans across different countries and continents [42]. Figure 1 shows the map of the city of Cape Town. According to the results of the most recent census conducted by Statistics South Africa [43], the majority of the population are coloureds (42.4%), followed by black Africans (38.6%), white Africans (15.7%%), and finally a small proportion of Asian or Indian communities (1.4%) and others (1.9%). Past evidence reveals that the coloured population has been falling over time, while the black population increased [44]. For example, the 2011 census found that the share of coloured people in the Western Cape fell from 54% in 2001 to 49.6% in 2011 while the white population dropped from 18.4% to 16%. The black population increased from 26.7% to 33.4% while the Indian population grew slightly to 1.1%. Growth in the black African population is caused by rural–urban migration and foreigners coming in from other African countries [43].

The economy of the city is based on tourism and wine farming which is also complemented by a vibrant retail industry of supermarkets and restaurants [45]. Most supermarkets combine both grocery and fast food sections to cater for the ever-growing market. An increase in the number of women recruited in the job market coupled with an increase in the number of consumers who prefer to eat fast foods signalled an increase in the demand for consumer goods and services and growth in the retail sector [46].

While Cape Town is not an exception, economic growth and urbanisation have contributed significantly to environmental pollution and the degradation of ecosystems in South African cities as firms and households seek to maximize their objectives. Like most cities in South Africa, the city of Cape Town has also experienced significant growth in informal settlements, which has also contributed to waste generation in the informal sector [47]. The occurrence of the informal sector, poverty and inequality contributes to the

complexity of implementing the green city philosophy in Africa. Even though the informal sector contributes a fair share of environmental pollution, it is difficult to instil discipline and sustainable behaviour in poor communities due to lawlessness and lack of infrastructure. Policymakers face a trade-off in terms of choices between policies that favour environmental sustainability and rehabilitation and policies that contribute to poverty reduction [2].

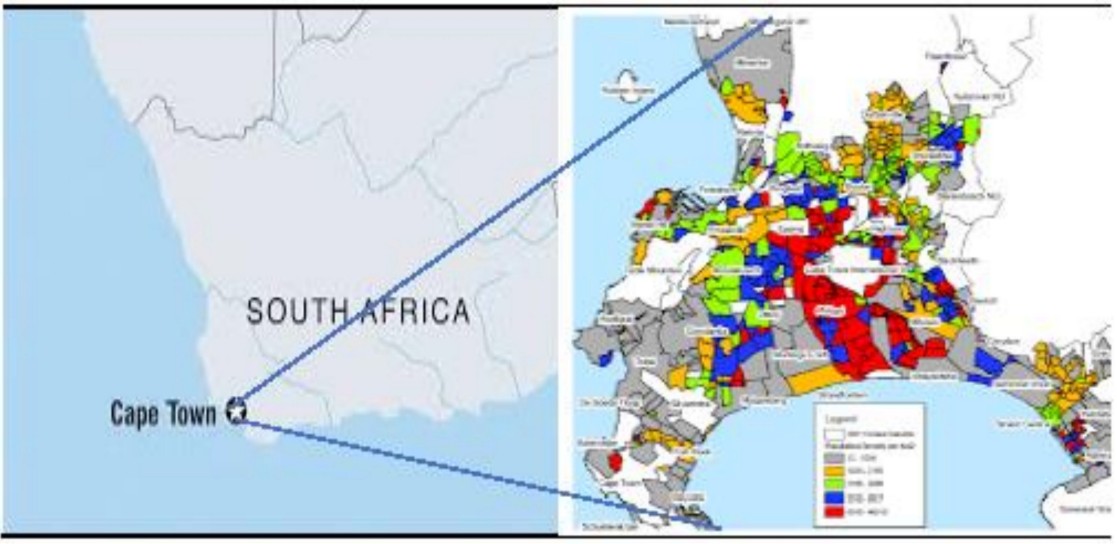

**Figure 1.** Map of the city of Cape Town (Source: Google Maps).

The retail business sector generates a significant amount of solid waste (e.g., plastics, glass, paper, cardboard) and organic waste (e.g., vegetables, meat, cooking oil and grey water). Most of the waste will eventually end up in landfill, on the streets, blocking drainage systems or polluting the oceans, while a small proportion is recycled or reused [48]. There is a call for the retail business sector to adopt sustainable behaviour to reduce the amount of pollution generated. Different strategies are available, and these include waste minimization or reduction, reuse, recycling, and recovery.

## 4. Research Methods

### 4.1. Justification of Choosing Choice Experiments

Stated and revealed preference methods are used in the literature to estimate the value of a product whose price is distorted due to market failure or in some cases where the market is totally absent [49]. Even if a market for a product exists, the market value may differ from the consumers' valuation of the attributes [18]. Theoretically and depending on how we define the product, we argue that plastic bags fit into one or two out of these three categories where these two methods apply since they pose externalities to society which are not internalized by the consumer [50]. A big challenge with the revealed preference in modelling a product with an externality is that it depends on the consumer's behaviour on the market which might not take into consideration external costs to the society and other unobservable factors during a purchase such as environmental pollution in the case of plastic bags [49]. The method only captures the market price of the shopping bag which is inadequate since it does not provide a true reflection of what is happening on the ground.

Under the stated preference methods, the contingent valuation method (CVM) and CEs are normally used to elicit the consumer's willingness to pay for a product. While the revealed preference method is based on the actual consumer behaviour on the market, the stated preference approach allows the consumer to think about the product in terms of WTP either as a whole as in CVM or by the disintegrating price of a product according to its characteristics as in CE [51]. Either way, both the CVM and CE allow the customer to think about those factors that are ordinarily not included in the market price [18].

Although sharing the same theoretical foundation as the CVM, the CE approach focuses on respondent preferences regarding the attributes of the scenarios in the design, rather than on specific scenarios [18,27,52].

In this study, we choose to use a CE since our objective is to identify the attributes of shopping bags that matter the most in the eyes of the consumers by eliciting their values. This information becomes imperative from a policy point of view as consumers are able to communicate their needs and wants, which in turn can be integrated into future policy design. Consumer choice in terms of packaging is purely an economic one since they weigh the costs and benefits of using one type over another [53]. Therefore, we can think of the attributes as conveying some kind of benefits to the consumer which lead to satisfaction.

As opposed to CVM which estimates the aggregate value that respondents place on a product when viewed as a whole, the CE estimates disintegrate the value of a product based on its attributes. CE borrows from Lancaster theory, which states that consumers think about a product in terms of its attributes, i.e., they are able to disaggregate a product into its attributes and value them separately and put individual values together to come up with total value. Lancaster theory should only hold if all the relevant attributes are known and aggregated. We ordinarily only include a few of those attributes in a CE to avoid cognitive burden. With this exercise, we merely try to understand how people's valuation of a good changes as they receive more information about its attributes. Based on the theory, we assume that people make better decisions when they have more information about a product.

### 4.2. Theoretical Underpinnings of the Model

The theoretical basis of CEs hinges on the characteristic of goods theory developed by [54] and random utility theory initially developed by [55] and popularized by [56] as its building blocks. The former theory states that people derive utility from the attributes of a commodity in addition to the mere consumption of the physical units of a good while the latter suggests that by observing a consumer choice, we cannot tell all the predictors of their utility [18]. A detailed discussion of the conceptual framework and underpinnings of the choice experiment approach in terms of an individual's decision-making and choice processes is provided by [57]. In principle, respondents are asked to choose the alternative they would prefer. According to [56] we can decompose the utility of consumer $i$ from alternative $j$, $V_{ij}$ into observable, $U_{ij}$ and unobservable $\varepsilon_{ij}$ components, i.e.,

$$V_{ij} = U_{ij}(A_j, X_i) + \varepsilon_{ij}(A_j, X_i) \tag{1}$$

where both components consist of the shopping bag attributes ($A_j$) and individual characteristics ($X_i$) of respondents. Analysis of SP data should allow for both observed and unobserved preference heterogeneity and should consider the relevance of this heterogeneity for the use of study results to support decision-making [24]. The consumer will only choose alternative $k$ over another one $j$ from a set $S$ if they derive a higher utility from $k$ compared to $j$. Alternative $k$ is chosen over alternative $j$, if $V_{iK} > V_{ij}$. The probability of a consumer choosing alternative $k$ over $j$ all comprising of a set $S$ can be expressed as:

$$P(k|S) = P[V_{iK} > U_{ij}] \ \forall k \ \neq \ j = P[(V_{iK} - V_{ij}) > \varepsilon ik - \varepsilon_{ij})] \ \forall k \neq j \tag{2}$$

Following the literature, we begin by estimating the utility with the conditional logit model (CLM), which assumes that each $\varepsilon_{ij}$ is independently and identically distributed (IID) with Weibull distribution, and consistent with the independence of an irrelevant alternative (IIA) [18,27,58]. The implication here is that the probability of choosing between options is not affected by other alternatives. Ref. [55] shows that a conditional logit model can be used to analyse the consumer choice with the attributes of the good or service acting as the predictors, and a ratio of the coefficients of attributes and prices used to recover the

marginal willingness to pay for an attribute. The CLM the probability of an individual $i$ choosing an alternative $j$ can be estimated by Equation (1) with the following general form:

$$P_{ij} = \frac{\exp\left(U\left(A_{ij}, X_i\right)\right)}{\sum_{k \in C} \exp\left(U\left(A_{ik}, X_i\right)\right)} \tag{3}$$

In other words, the difference in the systematic utility of alternative $k$ and $j$ exceeds the difference in the random utility of alternative $k$ and $j$. The difference in the observed utility is attributed to the difference in the attributes between alternative $k$ and $j$. The observable part is defined as a function of the attributes of the alternative and those of the respondent. From Equation (3) the conditional indirect utility function is estimated by Equation (4), which assumes a linear specification as follows:

$$U_{ik} = \beta_i A_{ik} + \delta X_i = \beta + \beta_1 A_1 + \beta_2 A_2 + \ldots + \beta_n A_n + \delta_1 X_1 + \delta_2 X_2 + \ldots + \delta_m X_m \tag{4}$$

where $\beta$ represents the alternative specific constant (ASC). However, if the CLM model violates the IIA property the model is likely to produce biased estimates because it assumes that all individuals have similar preferences. Furthermore, socio-economic and other varying characteristics are likely to be heterogeneous [59,60]. After running the CLM, we perform the Hausman test to see whether the IIA property is indeed violated.

If the CLM violates the IIA, then we can estimate an alternative model such as the RPL model which does not depend on this assumption [18,27] estimated an RPL model to assess the value of recreational ecosystem services in the case of urban parks in Dar es Salaam, while [18] used the same econometric technique to model preferences for CBNRM attributes. The advantage of the random parameter logit (RPL) model is that it relaxes the independence of irrelevant alternatives (IIA) assumptions and assumes continuous preference heterogeneity [55]. Under this model, the utility for individual $i$ from choosing alternative $j$ is shown in Equation (5), which is a modification of the utility function in Equation (1).

$$V_{ij} = U\left(A_j(\beta + \tau_i), X_i\right) + \varepsilon\left(A_j, X_i\right) \tag{5}$$

Under the RPL model, preference heterogeneity is allowed to vary across individuals by $\tau_i$ as a result of individual characteristics. Besides relaxing the IIA assumption, this is another important property of the RPL model which differentiates it from the CLM. In this case, the probability of an individual $i$ choosing option $j$ while accounting for preference heterogeneity is represented by Equation (6):

$$P_{ij} = \frac{\exp\left(U\left(A_j(\beta + \tau_i), X_i\right)\right)}{\sum_{k \in C} \exp\left(U\left(A_k(\beta + \tau_i), X_i\right)\right)} \tag{6}$$

In both the CLM and PRL models, parameters are specified as normally distributed, and the cost parameter is assumed to be fixed, while the rest of the parameters are randomly distributed. We then equate the utility levels from the different choice model specifications, i.e., CLM and RPL models, where we solve for price by obtaining the compensating surplus measure which is the individual's willingness to pay [61,62]. From the literature, the compensation variation ($CV$) is commonly used as a measure of welfare. The welfare changes experienced from changes in the level of an attribute can be computed using the estimated coefficients from either the CLM or RPL model as follows:

$$CV = \lambda^{-1} ln \left\{ \frac{\sum_{j \in C} \exp\left(U_j^1\right)}{\sum_{j \in C} \exp\left(U_j^0\right)} \right\} \tag{7}$$

where $\lambda$ represents the marginal utility of income, i.e., the estimated cost attribute from the choice experiments model. The utility functions of any individual before and after the changes in the urban agriculture programme are represented by $U_j^1$ and $U_j^0$, respectively.

From here, the marginal willingness to pay (WTP) for a change in any of the urban agriculture attributes is calculated as a ratio of: $-\beta_z/\lambda$, where $\beta_z$ is the estimated coefficient of zth choice attribute. The WTP estimates are calculated using the Wald (Delta method).

Apart from the CLM and RPL models, latent class analysis (LCA) is another econometric technique that is commonly used in the environmental economics literature to model preferences [27,58]. The advantage of using LCA is that we can recover unobserved classes in the data. The underlying argument supported by economic stylized facts is that consumers who belong to the same class (such as education, income) tend to have the same behavioural patterns [63,64]. LCA is then used to account for class heterogeneity in addition to preference heterogeneity while at the same time releasing the restrictive assumption of the IID of error terms [58]. Our objective of this analysis is to account for preference heterogeneity only as opposed to class heterogeneity, which makes the CLM sufficient for our analysis.

### 4.3. A Conceptual Framework for the Adoption of Sustainable Behaviour in a Circular Economy

Based on the theory discussed in Section 4.2, we adopt a conceptual framework based on the circular economy shown in Figure 2. A circular economy is a model of production and consumption, which involves reusing, repairing, refurbishing and recycling existing materials and products for as long as possible [40]. In contradistinction to the traditional linear economy, a circular economy tackles global challenges such as plastic pollution by emphasizing three base principles of the model during the design and implementation phase. The three principles required for the transformation to a circular economy are: eliminating waste and pollution, circulating products and materials, and the regeneration of nature.

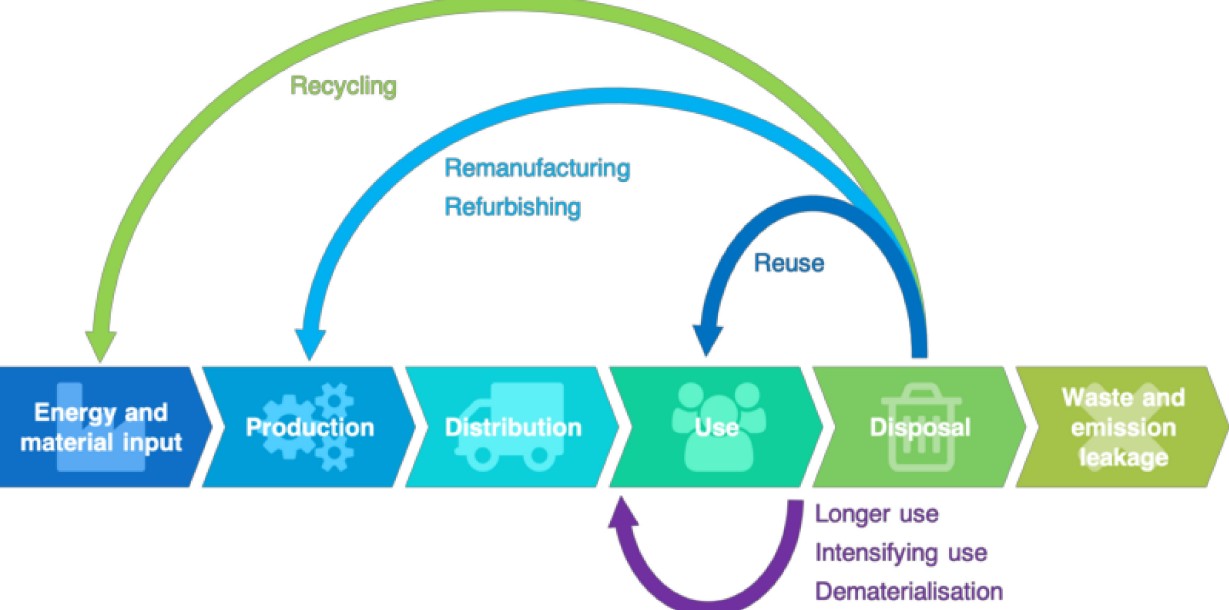

**Figure 2.** An illustration of the circular economy concept. Source: [65].

In a circular economy, there are different agents each contributing to the economic activities as shown in the model in Figure 2. This study deals with the mechanisms through which consumer behaviour can be influenced so that the circular economy model can be operationalized. For example, consumers can participate in recycling schemes, they can reuse or participate in upcycling of plastic bags. Without the consumers, the circular economy would not be as effective as it would be with their participation.

*4.4. Sampling and Data*

We randomly selected a sample batch of respondents in Cape Town from within the panel; thereafter, we used the randomized function to paste the survey links randomly alongside each email address within the sample batch. An equal number of respondents will be invited to each link. We had 4 survey links each representing a different scenario. The sample batch was selected first and then the 4 links were randomly added. Each link had an equal number of respondents invited. We created a sample batch of 1000 respondents and create 250 links per survey or scenario. The links were randomized and randomly pasted within the 1000 batches of the sample alongside each respondent. Survey invitations were sent out in small batches until we reached targets (please see Table 3).

**Table 3.** Sample characteristics.

| Location | Freq. | Percent | Cum. |
|---|---|---|---|
| *Proportion of very high-income areas* | | | |
| Constantia | 7 | 2.8 | 2.8 |
| Camps Bay | 35 | 14.0 | 16.0 |
| *Proportion of high-income areas* | | | |
| Claremont | 35 | 14.0 | 30.8 |
| Newlands | 14 | 5.6 | 36,4 |
| Mowbray | 8 | 3.2 | 39.6 |
| Milnerton | 14 | 5.6 | 45,2 |
| *Proportion of middle-income areas* | | | |
| Parklands | 14 | 5.6 | 50.8 |
| Goodwood | 40 | 16.0 | 66.8 |
| Maitland | 9 | 3.6 | 70.4 |
| *Proportion of low-income areas* | | | |
| Khayelitsha | 57 | 22.8 | 93.2 |
| Da Noon | 2 | 0.8 | 94.0 |
| Joe Slovo Park | 1 | 0.4 | 94.4 |
| Nyanga | 14 | 5.6 | 100.0 |
| **Total** | **250** | **100** | |

Source: Survey (2021).

The survey was administered in November 2021. To ensure compliance with ethical standards, the research instrument was initially assessed at the departmental level through a rigorous peer review process followed by the ethics committee in the Faculty of Business and Management Science at the Cape Peninsula University of Technology. Based on the evaluation and recommendation from the Faculty Ethics Committee, ethics approval or clearance was then issued by the university. This information was conveyed to the respondents in the introduction section of the survey instrument before asking for their consent. The respondents were initially asked about their awareness of sources of pollution, knowledge, use, preference, and reasons for their choice of the different types of shopping bags available on the market in the city of Cape Town. While the sampling was random, only those with access to email were sampled. The respondents were first made aware of the study through the following statement:

> *The purpose of this research study is to understand consumer awareness of plastic pollution, use of non-durable and durable plastic packaging and preference for shopping bag attributes by respondents in the city of Cape Town. You have been randomly chosen to complete this online survey. South Africa is one of several African countries studying how to reduce the consumption of plastic bags. The answers you and others give in the survey will provide empirical information to the retail industry and policymakers in the government that will be used to establish better pricing policies and programmes for plastic bad use. A plastic levy will be used as a vehicle to collect revenues. Do you give your concert to participate in this survey?*

A key feature of this message is that it is designed to embed some consequentiality, giving the respondents some sense that the results of the survey might potentially influence an outcome they care about [52,66]. Furthermore, the survey respondents were informed that the plastic levy is reviewed periodically as part of government policy and this is through a consultative process with different stakeholders including customers. This should enable us to reduce hypothetical bias to a degree [52]. After a series of questions to ascertain awareness of plastic pollution, knowledge, use and preferences for different shopping on the market, respondents were then presented with the following hypothetical CE question designed to elicit their WTP:

> *The use of non-durable or single-use plastic bags is on the increase despite availability of alternative shopping bags. It is still not clear how customers make decisions or the reasons why consumers choose the former over the latter. What is known is that most of the plastic packaging used on the market are pushed by retail outlets without consumers being aware of their buying behaviour since they are very cheap. An intervention that could incentivize consumers would be to subsidize durable and expensive packaging to increase uptake so that consumers can reduce consumption of non-durable plastic packaging by consumers.*

Both the introduction to the survey and the CE question were carefully crafted to minimize bias by avoiding terms and concepts that might lead the respondents such as plastic pollution. We wanted the respondents to behave as if they are faced with a real-life decision scenario at the point of purchase, so we introduced the CE question much earlier than other sections addressing pollution. After reading this introduction, the respondents were initially asked if they supported such a policy if instituted by the government. The respondents were then presented with four choice cards with three different options (option 1, option 2 and status quo) and asked to choose among the three alternatives. After presenting the CE experiment, questions about the socio-economic characteristics of the respondent concluded the survey.

## 5. Results

### 5.1. Descriptive Statistics

Table 4 presents the characteristics of the sampled respondents. The proportion of male respondents in the sample is 67.6%, 54.0% were household heads and 66.0% of the household heads were employed. Considering the level of education in the sample, our results show that 36.4% of the respondents had high school or lower, 28.4% had certificates or diplomas, 26.0% had undergraduate degrees, and a negligible proportion had postgraduate qualifications. Considering the respondents' profiles by race, we observe that 36.4% were coloured, 32.4% were white, 28.4% were black Africans and 2.8% were from Asian communities. The average age of the respondent is 29.8 years, the household size is 3.4 members per household and the total household income is R18,034 with a standard division of R14,429.

**Table 4.** Characteristics of the respondent.

| Variable | | Obs. | Mean | Std. Dev. | Min | Max |
|---|---|---|---|---|---|---|
| Gender | | 250 | 0.676 | 0.469 | 0 | 1 |
| Head (Is respondent the household head?) | | 250 | 0.540 | 0.499 | 0 | 1 |
| Employed (employment status of head) | | 250 | 0.660 | 0.475 | 0 | 1 |
| Education | High school/lower | 250 | 0.364 | 0.482 | 0 | 1 |
| | Certificate/diploma | 250 | 0.284 | 0.452 | 0 | 1 |
| | Undergraduate | 250 | 0.260 | 0.440 | 0 | 1 |
| | Postgraduate | 250 | 0.092 | 0.290 | 0 | 1 |
| Race | Black | 250 | 0.284 | 0.452 | 0 | 1 |
| | White | 250 | 0.324 | 0.469 | 0 | 1 |
| | Coloured | 250 | 0.364 | 0.482 | 0 | 1 |

**Table 4.** *Cont.*

| Variable | | Obs. | Mean | Std. Dev. | Min | Max |
|---|---|---|---|---|---|---|
| | Asian | 250 | 0.028 | 0.165 | 0 | 1 |
| Sector | Retail | 250 | 0.196 | 0.398 | 0 | 1 |
| | NGO | 250 | 0.168 | 0.375 | 0 | 1 |
| | Education | 250 | 0.260 | 0.440 | 0 | 1 |
| | Health | 250 | 0.212 | 0.410 | 0 | 1 |
| | Environment | 250 | 0.572 | 0.824 | 0 | 1 |
| Age of the respondent | | 250 | 29.80 | 8.771 | 18 | 67 |
| Household size | | 250 | 3.420 | 2.555 | 0 | 16 |
| Total household income | | 250 | 18,034 | 14,429 | 3500 | 45,500 |

Source: Survey (2021).

Table 5 shows the respondent's awareness of plastic pollution and preference for different shopping bags on the market. Our results show that 97.2% are aware of plastic pollution. An insertion demonstrating how respondents feel about plastic pollution during an FGD is provided in Text Box 1. When asked whether they would prefer a different shopping bag other than single-use plastic bags, 78.4% indicated their preference for alternative packaging. Interestingly, 87.8% reported that they prefer shopping bags that are made of cloth, followed by 48.0% who indicated their preference for paper bags, 46.9% for taxi bags and finally 38.8% still prefer single-use plastic bags. Preference for different shopping bags is based on attributes such as reusability (95.9%), durability (91.8%) and attractiveness (82.7%). Cost was highlighted by 56.6% of the respondents as a major impediment to the use of alternative packaging. Information gathered through FGDs and key informant interviews indicates that most shoppers choose single-use plastic bags because they are cheap as compared to alternative shopping bags.

**Table 5.** Preference for different packaging.

| Variable | Obs. | Mean | Std. | Min | Max |
|---|---|---|---|---|---|
| Awareness of plastic pollution | 250 | 0.972 | 0.165 | 0 | 1 |
| Would you prefer a different type of packaging or bag? | 250 | 0.784 | 0.412 | 0 | 1 |
| Which one would you prefer? | | | | | |
| Single-use plastic bags | 196 | 0.388 | 0.488 | 0 | 1 |
| Paper bag | 196 | 0.480 | 0.501 | 0 | 1 |
| Taxi bag | 196 | 0.469 | 0.500 | 0 | 1 |
| Bag made of cloth | 196 | 0.878 | 0.329 | 0 | 1 |
| Why do you prefer this type of packaging? | | | | | |
| It is durable | 196 | 0.918 | 0.275 | 0 | 1 |
| It is attractive | 196 | 0.827 | 0.380 | 0 | 1 |
| It is reusable | 196 | 0.959 | 0.198 | 0 | 1 |
| Is cost the major reason why you are not using this packaging? | 196 | 0.566 | 0.497 | 0 | 1 |
| What is the highest price that you are willing to pay for a non-durable plastic bag? | 250 | 2.92 | 4.21 | 0 | 35 |
| What is the minimum price that you would consider for alternative packaging? | 250 | 7.365 | 12.05 | 0 | 120 |

Source: Survey (2021).

**Box 1.** Insertion demonstrating how respondents feel about plastic pollution during an FGD.

> *Plastic pollution is like a deadly disease which has entered our community. We drink, eat and sleep with pollution. There is plastic pollution everywhere we look and go. People do not have morals and no body seem to care. What are we teaching our children? What type of a world will they inherit from us? Are they going to be proud and say that this is the inheritance that the generation before left for us? No one knows how this will end and no one has got a solution to this problem—not even the city authorities. Source: FGDs (2021).*

The maximum price that respondents are willing to pay for single-use plastic bags is R2.92, while the minimum price that respondents are willing to pay for durable packaging is R7.37 on average with a standard deviation of R4.21 and R12.05, respectively. The proportion of respondents with a choke price greater than the average (R2.92) is less than 30.0% of the sample. Out of those who will consider available alternatives, 88.5% indicated a preference for carrying bags made of cloth, 64.6% prefer taxi bags and 52.6% would consider paper bags.

*5.2. Model estimation Results*

Following standard practice in the literature, our analysis considered the effects of undesirable response anomalies such as protest or outlier responses [18,24,27]. In this experiment, protest responses are those where respondents do not genuinely choose the status quo for various reasons. We asked appropriate questions to identify these protestors. Based on the results of this analysis, no protestors were identified. To control for the difference between status quo and non-status quo alternatives, we included a dummy equal to one for status quo and zero for the other options. This was also because the two alternatives other than the status quo had the same sign and were almost equal in magnitude. The inclusion of the dummy also measures some propensity to choose a zero-cost option or status quo. Table 6 shows the frequency with which each alternative was selected. We use these results as an indication of the presence of the status quo bias. The status quo bias is small approximately 15.8%. This shows that a small proportion of the sampled respondents would prefer the two alternative options to the status quo. Based on the FGDs, more preference would be given to multi-use shopping bags if they were affordable. Our results also show that Option B is more preferred to Option A suggesting a lack of balance between the two alternatives.

**Table 6.** Choice frequency.

| Option | Frequency | Percentage |
|--------|-----------|------------|
| Option A | 333 | 33.3 |
| Option B | 509 | 50.9 |
| Option C | 158 | 15.8 |
| **Total** | **1000** | **100.00** |

Source: Survey (2021).

We begin by testing whether the CLM assumption of the Independence of Irrelevant Alternative (IIA) property holds using the Hausman test report in Table 7. According to the Hausman test, the IIA assumption is significantly violated if any of the alternatives are dropped from the choice set. The results indicate that the IIA assumption is significantly violated, suggesting that the use of CLM might yield biased results. For this reason, we will only interpret the result of the RPL model.

**Table 7.** Independence of the irrelevant alternative (IIA) assumption test.

| Alternative Dropped | Chi-Square ($\chi^2$) | Probability | Comment |
|--------|-----------|------------|--------|
| Choice 1 | 25.75 | 0.001 | No violation |
| Choice 2 | 3.36 | 0.422 | Violation |
| Status-quo | 1.80 | 0.880 | Violation |

Source: Survey (2021).

The CLM and RPL estimation results are presented in Table 8. The results of both models seem to agree with each other considering the sign only but differ when it comes to the level of significance and magnitude of the coefficients. As already alluded to earlier, we will interpret the results of the later model since the IIA assumption has been violated

which renders the results of the CLM unfit for this analysis. Our results show that the RPL model is highly significant compared to the CLM. Furthermore, the RPL model included interaction between the attributes of shopping bags and location to capture wealth effects in the analysis. Initially, we estimate the RPL model without interaction terms and then we re-estimate the model with interaction terms. As expected, the cost coefficient is negative and significant indicating that utility diminishes as the cost of a shopping bag increases. The negative and significant signs on the ASC coefficient suggest that the bias could be negligible which supports earlier findings in Table 6 and FGDs that preference for the status quo is insignificant relative to other alternative options. Please note that through the analysis reference is made to the baseline scenario which is the status quo against which the results are compared.

**Table 8.** Regression model results.

| Variable | CLM | | RPL | | RPL with Interaction | |
|---|---|---|---|---|---|---|
| | Coef. | Std. Err. | Coef. | Std. Err. | Coef. | Std. Err. |
| | *Random parameters* | | | | | |
| small_size | 0.0286 ** | 0.157 | 0.0385 ** | 0.287 | 0.0474 ** | 0.376 |
| medium_size | −0.0459 *** | 0.168 | −0.0620 *** | 0.135 | 0.0511 *** | 0.224 |
| large size | −0.00673 * | 0.260 | −0.00191 *** | 0.211 | −0.00282 *** | 0.302 |
| Recyclable | 0.0257 * | 0.098 | 0.0206 *** | 0.351 | 0.0315 *** | 0.262 |
| Reusable | 0.0113 * | 0.094 | 0.0552 ** | 0.363 | 0.0461 ** | 0.272 |
| Durable | 0.0202 *** | 0.092 | 0.0247 ** | 0.0448 | 0.0338 ** | 0.0357 |
| Style | 0.0307 * | 0.098 | 0.0356 *** | 0.0523 | 0.0437 *** | 0.0432 |
| small_size × lowincomearea | | | | | 0.00378 *** | 0.0477 |
| medium_size × lowincomearea | | | | | 0.00994 | 0.0383 |
| large_size × lowincomearea | | | | | −0.00128 *** | 0.0368 |
| recycle_size × lowincomearea | | | | | 0.00120 | 0.0355 |
| reuse_size × lowincomearea | | | | | 0.00139 *** | 0.0369 |
| durable_size × lowincomearea | | | | | 0.00123 | 0.00871 |
| style_size × lowincomearea | | | | | 0.00109 | 0.0147 |
| small_size × mediumincomearea | | | | | −0.00578 | 0.0386 |
| medium_size × mediumincomearea | | | | | 0.00842 * | 0.0185 |
| large_size × mediumincomearea | | | | | −0.00448 ** | 0.0259 |
| recycle_size × mediumincomearea | | | | | 0.00341 | 0.0246 |
| reuse_size × mediumincomearea | | | | | 0.00152 ** | 0.0478 |
| durable_size × mediumincomearea | | | | | 0.00345 * | 0.00871 |
| style_size × mediumincomearea | | | | | 0.00210 | 0.0147 |
| small_size × highincomearea | | | | | −0.00469 ** | 0.0477 |
| medium_size × highincomearea | | | | | 0.00842 *** | 0.0383 |
| large_size × highincomearea | | | | | 0.00537 | 0.0368 |
| recycle_size × highincomearea | | | | | 0.00230 ** | 0.0355 |
| reuse_size × highincomearea | | | | | 0.00241 * | 0.0369 |
| durable_size × highincomearea | | | | | 0.00345 ** | 0.00871 |
| style_size × highincomearea | | | | | 0.00210 *** | 0.0147 |
| | *Nonrandom parameters* | | | | | |
| ce_cost | −0.0187 *** | 0.00556 | −0.0174 *** | 0.00536 | −0.0165 *** | 0.00645 |
| Asc | −0.226 *** | 0.120 | −0.0269 *** | 0.283 | −0.0278 *** | 0.392 |
| Number of obs. | 3000 | | 3000 | | 3000 | |
| LR chi$^2$ (8) | 14.38 | | 18.39 | | 33.48 | |
| Prob > chi$^2$ | 0.0731 | | 0.0015 | | 0.000 | |
| Log likelihood | −1749 | | −1749 | | −2658 | |

Source: Survey (2021). * sig at 10% ** sig at 5% *** sig at 1%.

Our results show that all the attributes included in the model carry positive signs except for large-sized shopping bags. Initially, the coefficient for medium-sized shopping bags carries a negative sign in the model without interactions, but with the inclusion of interaction terms, it becomes positive. The sample respondents revealed their preference for

small- and medium-sized shopping bags that are recyclable, reusable, durable and styled compared to the status quo. Consistent with the FGD results, less preference is given to very large shopping bags compared to the status quo. When we disaggregate the analysis by location, we observe that small-sized shopping bags are preferable relative to the status quo in low-income areas, while medium-sized shopping bags are more preferred compared to the status quo in medium- and high-income areas. Small-sized shopping bags are less preferred in high-income areas. However, key informant interviews revealed high uptake of both medium and small-sized shopping bags in all areas. Although reusable shopping bags are preferred relative to the status quo across all income areas, the significance level decreases as we move from low to high-income areas. Large-sized shopping bags are less preferred compared to the status quo in low- and medium-income areas. Durable shopping bags are preferred compared to the status quo in medium and high-income areas, while styled and branded shopping bags are preferred compared to the status quo in high-income areas only.

### 5.3. WTP Estimates for the CLM and RPL Model

According to the RPL model, the highest WTP is associated with the medium-sized shopping bag (R3.76), followed by a shopping bag that is reusable (R3.35), then a shopping bag that is styled (R2.16), small-sized shopping bag (R1.74), a durable shopping bag (R1.50) and finally a recyclable shopping bag (R1.25). Both FGDs and key informant interviews identified reusability as an important attribute that consumers care about. We can arrive at a price of a shopping bag by aggregating the amounts associated with each attribute. For example, if we consider a small shopping bag that is recyclable, the WTP is R2.99 which is equivalent to the maximum price that the respondent is offering for a single-use plastic bag. Taking the first and second highest WTP associated (i.e., medium-sized and reusable shopping bags), the aggregated figure amounts to R7.11 per unit which is at least as high as the minimum price that respondents would consider for alternative packaging. Alternatively, if we consider the medium-sized shopping bag and all the attributes of the shopping bag, we arrive at a maximum WTP of R12.01 per bag (Table 9).

**Table 9.** WTP estimates for the CLM and RPL model.

| Attributes | WTP | |
|---|---|---|
| | **CL Model** | **RPL Model** |
| small_size | 1.53 | 1.74 |
| medium_size | 2.45 | 3.76 |
| large_size | 0.36 | 0.16 |
| Recyclable | 1.37 | 1.25 |
| Reusable | 0.60 | 3.35 |
| Durable | 1.08 | 1.50 |
| Style | 1.64 | 2.16 |

Source: Survey (2021).

Table 10 shows discrepancies between the WTP and ranking by the respondents themselves of the attributes. The respondents were asked to rank the attributes of shopping bags before the CE as a check-up question. The WTP and raking agree on four attributes (i.e., small size, medium size, reusable and durable shopping bags) out of seven attributes and differ on three attributes (namely style, recyclable and large-sized shopping bags).

**Table 10.** Ranking of attributes.

| | WTP | | Actual Ranking | |
|---|---|---|---|---|
| Medium-sized shopping bag | R3.76 | 1 | 37.4 | 1 |
| Reusable shopping bag | R3.35 | **2** | 18.7 | **2** |
| Styled shopping bags | R2.16 | 3 | 3.7 | 6 |
| Small sized shopping | R1.74 | 4 | 12.5 | 4 |

**Table 10.** *Cont.*

|  | WTP | | Actual Ranking | |
|---|---|---|---|---|
| Durable shopping bag | R1.50 | 5 | 8.0 | 5 |
| Recyclable shopping bag | R1.25 | 6 | 17.8 | 3 |
| Large-sized shopping bag | R0.36 | 7 | 1.9 | 7 |

Source: Survey (2021).

## 6. Discussion

Our sampled respondents are consistent with young, educated, and working-class respondents with access to the internet. In present-day society, these are either the decision makers or breadwinners in poor and medium-income households in most developing countries [42]. It is also not surprising that most of the respondents who either participated in FGDs or reacted to this survey are interested in environmental issues given that plastic pollution is increasingly becoming an issue in many societies and most jobs address this issue. Previous studies also indicated a very high level of awareness of plastic pollution in African cities [1,20]. As a result, there might be self-section bias which we could not solve in this study since most of the respondents who responded to this survey either work in the environment sector or have some sort of environmental education background from school and are aware of plastic pollution. In designing our choice experiment, we made sure to avoid linking plastic bags with environmental pollution at the beginning which could bias our results. The choice experiment was administered first before respondents were asked about their awareness of plastic pollution. As a result, the decision that respondents made during the survey is similar to a real-life scenario they face at the point of purchase. Because of this, we do not believe that our results would differ if another random sample of respondents which includes those without access to the internet was drawn.

Evidence shows that both low- and medium-income households are the biggest consumers of single-use plastic bags, partly because they are very cheap and partly because of a lack of awareness of either the magnitude of the problem of plastic pollution in their area or its consequences [4,9]). The latter is a stylized fact in the literature which has not been subjected to rigorous econometric testing in developing countries [8]. Our survey results show high awareness levels of plastic pollution in Cape Town across different population sub-groups. This result is also corroborated by qualitative information gathered through FGDs and key informant interviews. Given a very high level of awareness of plastic pollution, most respondents indicated a preference for alternative shopping bags which is also consistent with previous studies [23,30]. Although most of these studies are from first-world countries, there is increasing awareness of plastic pollution and a need to change to alternative packaging [9]. Most of the respondents in this study indicate a preference for shopping bags made of cloth because they are durable, attractive, and reusable. Why then does rhetoric deviate from the actual behaviour of the respondent? According to survey results, key informant interviews and FGDs, the answer to this question might be associated with the cost of alternative shopping bags which act as a deterrent to most customers.

The maximum that respondents are WTP for single-use plastic bags (choke price) is several times higher than the current price that respondents are facing on the market, while the minimum they are WTP for alternative packaging, which we refer to as the price floor in our analysis, is way below the market price. On one hand, our results confirm another stylized fact from previous studies that the plastic levy could be suboptimal [5]. On the other hand, our results speak to the use of market-based policy instruments aiming to reduce the utilization of single-use plastic bags either through an increased plastic levy or increased consumption of multiuse packaging via a subsidy. This means that increasing the levy could act as a disincentive for single-use plastic bags while a subsidy acts as an incentive to increase the consumption of alternative shopping packaging [67]. An optimal outcome might be achieved if these two policy instruments are used in combination rather than in isolation [8]. However, it is also important to note that the benefits of combining these two policy instruments might not necessarily exceed the benefits of using either a levy

or subsidy in isolation [18]. A system where the joint effect of two or more interventions is dissipated is inefficient and is costly to society. The conditions under which this happens are not known with certainty since this depends on the context and might be subject to empirical investigation.

It is not surprising that the status quo bias is insignificant given that most respondents are aware of plastic pollution. Consistent with previous studies, our results show that the drive to reduce plastic pollution is present in all communities under consideration whether we consider low-, medium- or high-income areas. Consistent with previous studies, the results of this study demonstrate unhappiness with deteriorating environmental conditions as a result of plastic pollution [1,20]. According to FGDs and key informant interviews, residents in low-income areas identified plastic pollution as one of the main causes of blockages of the sewage and drainage system and flooding as its impacts. Evidence shows a reduced value of properties located near dumpsites and informal settlements where the challenge is most pronounced [68].

Consistent with what is happening on the ground, preference for small-sized shopping bags is associated with convenience in use and disposal [32]. The problem is that when customers make a purchase decision, they do not take into consideration convenience in use and the externality that is imposed on society when the shopping bag is final dispose-off [50]. It is debatable whether consumers make a joint decision about consumption and pollution at the point of purchase when they buy plastic bags given information constraints, particularly in developing countries. Given that customers are aware of the problem of plastic pollution, the type of information conveyed becomes imperative from a policy point of view to incentivize a reduction in plastic consumption [1]. The common narrative is that plastic pollution is associated with poor households or communities simply because plastic litter is dumped on the street, yet a huge amount of litter is channelled through the formal system which ends up in dump sites or the ocean [69]. Such contribution to plastic pollution is often neglected since it is unobservable or nonattributable to a single community. Therefore, rather than conveying the usual message about pollution, an awareness campaign could be tailored to suit the needs of different market segments.

Our results demonstrate a preference for both small- and medium-sized shopping bags relative to the status quo since they are more convenient for most users. Small- and medium-sized packaging do not only allow shoppers to disaggregate goods into small units but also according to other measures of convenience such as the danger posed by combing food stuff and chemicals [33]. FGDs revealed that most shoppers prefer to use small and medium-sized shopping bags together for more convenience. In this sense, small and medium-sized shopping bags actually act as complementary tools rather than substituting each other. Therefore, large shopping bags are less preferred since they do not offer these attributes. Furthermore, the use of different sizes of shopping bags makes it possible for customers without transport to strike a balance between short and long-distance travel on foot [9].

The fact that small-sized shopping bags are more preferable compared to the status quo in low-income areas confirms that poor households are more concerned with convenience in use since they carry the goods long distances, while medium-sized shopping bags are more preferred relative to the status quo in medium- and high-income areas since they have transport [1]. Small-sized shopping bags are less preferred compared to the status quo in high-income areas perhaps because the respondents are aware of plastic pollution or they own transport and hence need to save costs. Reusable shopping bags are preferred compared to the status quo across income areas. This is interesting because the motives could be different. For example, poor households are aiming to save costs, while nonpoor households are concerned about the environment [9,23]. Large-sized shopping bags are less preferred relative to the status quo in low- and medium-income areas since convenience is compromised, especially if the shopper travels long distances [70]. Large-sized shopping bags makes shopping bags heavy and difficult to handle [1]. Consistent with the literature, a preference for more durable shopping bags in medium and high-income areas could be

associated with environmental concerns [70]. It is not surprising that styled and branded shopping bags are preferred in high-income areas since evidence reveals that relatively wealthy customers respond to such marketing strategies as income increases [9].

Our results show that the highest WTP is associated with the medium-sized shopping bag confirming that customers prefer averages to extremes as suggested in microeconomic theory [71,72]. The second highest WTP is associated with reusable shopping bags, followed by styled shopping bags. According to FGDs, the least preferred or lowest WTP is associated with large-sized shopping bags since carrying the goods becomes heavy and difficult, especially if the customer does not have transport. Surprisingly, the WTP for recyclable shopping bags is also low even though this attribute is ranked high by respondents.

From a policy perspective, we are interested in the convergence between the WTP and the maximum price that respondents are willing to pay for a single-use plastic bag (choke price) or the minimum price for alternative durable shopping bags (price floor). From the characteristics of goods theory, respondents are able to disaggregate the price of a product according to its attribute and then aggregate the individual prices again to arrive at the price of a whole. By aggregation, we arrive at the following conclusions and prices of different types of shopping bags. The WTP of a small shopping bag that is recyclable is R2.99 which approaches the choke price for a single-use plastic bag of R2.92. Therefore, the optimal price of a recyclable is around the delta neighbourhood per unit. The choke price diverges from the market price of recyclable plastic bags whose value is currently pegged at R0.75 per unit. The difference between these two values can be imposed on the consumer as a levy so that the externality is internalized. Taking the first and second highest WTP associated (i.e., medium-sized and reusable shopping bags), the aggregated figure amounts to R7.11 per unit which is at least as high as the minimum price (R7.37) that respondents would consider for alternative packaging. Without other attributes factored in, this price (R7.11) might result in an inferior product since manufacturers are not taking into consideration the full range of attributes. Key informant interviews and FGDs revealed that a significant number of shopping bags that are found on the market are inferior as they break down easily. Furthermore, if we consider the medium-sized shopping bag and all the attributes of a shopping bag, we arrive at a maximum WTP of R12.01 per bag. Without loss in quality, the minimum price for a durable shopping bag of R7.37 is attainable if manufacturers are compensated for production costs. Policies such as a tax break or subsidies might assist in realizing such a significant reduction in the costs of durable shopping bags.

## 7. Conclusions

Single-use plastic bags are increasing becoming unpopular across the globe due to increasing plastic pollution which is threatening both terrestrial and marine ecosystems. The situation is even worse in developing countries and economies in transition such as South Africa where plastic consumption is fuelled by a combination of population growth and increasing household incomes resulting from the increased participation of women in the labour force and longer working hours. Interventions to reduce plastic consumption require objective information so that robust policies can be crafted to suit local conditions.

This study uses a choice experiment to elicit consumer preference for attributes of shopping bags from a sample of 250 consumers in Cape Town. Following the literature, we estimate two different models. First, we estimated the conditional logit model as the baseline model and perform appropriate tests to establish a model which fits our data. The Hausman–Mcfadden test was used to examine if the CLM violated the IIA assumption. Since the CLM violated the IIA assumption, we then proceeded to estimate the mix logit model or RPL model where the attributes interacted with the location.

Contrary to one stylized fact which paints poor people as being ignorant, unaware, and uninformed, we found awareness of plastic pollution to be very high among the surveyed respondents from low- to high-income areas. When asked about their preference for shopping bags, most respondents indicated that they would prefer durable shopping

bags made of cloth. The highest price that respondents are WTP for non-durable (single-use) plastic bags is R2.92 per bag, while the minimum price that they would consider for alternative packaging is R7.37 per unit. This shows that there is room for increasing the plastic levy to force customers to internalize the externality. All attributes included in the model carry positive signs, except for large-sized shopping bags, implying that their presence enhanced the consumer's utility. Large-sized plastic bags are less preferred compared to the status quo. Small-sized plastic bags are more preferred in low-income areas since they are relatively cheaper, while medium-sized packaging is more preferred in medium- and high-income areas.

According to the RPL model, the highest WTP is associated with the medium-sized shopping bag (R3.76), followed by a shopping bag that is reusable (R3.35), then a shopping bag that is styled (R2.16), a small-sized shopping bag (R1.74), a durable shopping bag (R1.50) and finally a recyclable shopping bag (R1.25). The low WTP value for recyclable shopping bags could be an indication that the respondents are not aware of recycling schemes. The WTP of a small-sized shopping bag that is recyclable is R2.99 which is equivalent to the maximum price that the respondent is offering for a single-use plastic bag (R2.92). Taking the first- and second-highest WTP associated (i.e., medium-sized and reusable shopping bag), the aggregated figure amounts to R7.11 per unit which is at least as high as the minimum price that respondents would consider for alternative packaging. Furthermore, if we consider the medium-sized shopping bag and all the attributes of the shopping bag, we arrive at a maximum WTP of R12.01 per bag.

Based on these results, the preferences of the sampled consumers in Cape Town do not tell us much about sustainable behaviour since the WTP values of durability and recyclability are low. Although not addressed in our study, the possibility of upcycling plastic bags as a sustainable solution to the disposal of shopping bags and contribution to the circular economy must not be underestimated as the growth of new ideas in this area can be promoted through awareness campaigns, learning, co-creation and competition to stimulate and reward innovation. Based on our results, we derive four policy recommendations.

- Our results call for a combination of policy instruments such as a subsidy on expensive durable and reusable shopping bags to increase demand while at the same time increasing the levy on single-use plastic bags to reduce demand.
- The message behind awareness campaigns and educational programmes should be carefully crafted to increase awareness of the benefits of upcycling and participation in recycling schemes rather than focusing only on plastic pollution and its negative effects.
- Educational materials through television programs and internet-based platforms or social media could be used to enhance upcycling of plastic bags.
- Policymakers can make use of the prices provided in this paper as an initial guide to the optimal value of the different types of plastic bags.

**Author Contributions:** Conceptualization, V.V.M.; Formal analysis, H.N.; Investigation, H.N.; Methodology, H.N.; Project administration, V.V.M.; Software, H.N.; Supervision, V.V.M. All authors have read and agreed to the published version of the manuscript.

**Funding:** This research received no external funding.

**Institutional Review Board Statement:** Not applicable.

**Informed Consent Statement:** Not applicable.

**Data Availability Statement:** Not applicable.

**Conflicts of Interest:** The authors declare no conflict of interest.

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
