# Peer review of "Consumer Preference for Attributes of Single-Use and Multi-Use Plastic Shopping Bags in Cape Town: A Choice Experiment Approach"

_sustainability, doi:10.3390/su141710887_

Round 1

Reviewer 1 Report

Good paper addressing the major pollution problem

The course of experiment is clear but can be imporved by being a little more brief in presenting the details, in the present form the paper seems overloaded.

The recommendations on policies have to be presented with the bullet-points - they are not very clear here. 

Very few new sources referenced in the literature  - the list needs to be updated with some sources  of 2020-2022

Author Response

Dear Reviewer 1

Thanking for suggesting improvements that we could make to our paper. We indeed apprecaition your valuable comments and we have incorporated them into the paper.

Please therefore find the attached document with all our responses to your suggestions.

Regards

Authors

Reviewer 2 Report

Thank you for the opportunity to review this paper. The author have put much effort in delivering the final manuscript and used proper methodology. There are some issues in this paper that needs to be taken into consideration: 

- The introduction part in the end should explain the research flow of the manuscript, 

- the main issue for me in the paper is the topic which is researched which as it is mainly contributes to policymakers or business owners but very little to the research and the the literature. I would recommend changing the research question and the title of this paper, the topic is interesting ( analyzing the impact of single use or multi use bags however it has to analyze maybe their impact in the environment......

Author Response

Dear Reviewer 2

Thanking for suggesting improvements that we could make to our paper. We indeed apprecaition your valuable comments and we have incorporated them into the paper.

Please therefore find the attached document with all our responses to your suggestions.

Regards

Authors

Reviewer 3 Report

1. Write WTP in full, in the abstract. 

2. There a lot of self citation and at times not necessary. Cite the views of other authors.

3. Upcycling should have been one of the attributes when making a decision to purchase a shopping bag. Upcoming is one of the drivers of the circular economy. 

4. I had wished the authors could have highlighted the circular economy in the theoretical framework on how their study will contribute to the attainment of the same.

5. Did the study obtain a research permit to safeguard ethical issues? This is not clear in the manuscript.

6. Did  all the 250 participants sampled consented to take part in the study?

7. How many respondents participated in the FGDs and key informant interviews? How were they sampled? Were the participants consent obtained before the interviews?

8. The results from the FDGs and key informant interviews are not pronounced except in Text box 1. Is that the only thing respondents were asked?

9. The conclusion should discuss the possibility of upcycling shopping bags as a sustainable solution to disposal of shopping bags and contribution to the circular economy. 

10. Mugobo and Ntuli 2021a and 2022a are over cited in the discussion section. Are there no other sources which can be cited? This makes the discussion biased towards the opinion of the two authors.

11. There are a few grammer and spelling errors that need to be corrected. 

Author Response

Dear Reviewer 3

Thanking for suggesting improvements that we could make to our paper. We indeed apprecaition your valuable comments and we have incorporated them into the paper.

Please therefore find the attached document with all our responses to your suggestions.

Regards

Authors

Round 2

Reviewer 2 Report

The authors have implemented all recommandations!